# Impact of Urbanization on Ecosystem Health: A Case Study in Zhuhai, China

**DOI:** 10.3390/ijerph16234717

**Published:** 2019-11-26

**Authors:** Nan Cui, Chen-Chieh Feng, Rui Han, Luo Guo

**Affiliations:** 1School of Life and Environmental Science, Minzu University of China, Beijing 100081, China; Cuinan1994@outlook.com (N.C.); 18301234@muc.edu.cn (R.H.); 2Centre for Spatial Analysis and Policy, School of Geography, University of Leeds; Leeds LS2 9JT, UK; 3Department of Geography, National University of Singapore, Singapore 117570, Singapore; chenchieh.feng@nus.edu.sg

**Keywords:** ecosystem health, remote sensing images, urbanization, spatial correlation, comprehensive indicators

## Abstract

The past decades have witnessed rapid urbanization around the world. This is particularly evident in Zhuhai City, given its status as one of the earliest special economic zones in China. After experiencing rapid urbanization for decades, the level of ecosystem health (ESH) in Zhuhai City has become a focus of attention. Assessments of urban ESH and spatial correlations between urbanization and ESH not only reveal the states of urban ecosystems and the extent to which urbanization affected these ecosystems, but also provide new insights into sustainable eco-environmental planning and resource management. In this study, we assessed the ESH of Zhuhai City using a selected set of natural, social and economic indicators. The data used include Landsat Thematic Mapper images and socio-economic data of 1999, 2005, 2009 and 2013. The results showed that the overall ESH value and ecosystem service function have been on the decline while Zhuhai City has continued to become more urbanized. The total ESH health level trended downward and the area ratio of weak and relatively weak health level increased significantly, while the areas of well and relatively well healthy state decreased since 1999. The spatial correlation analysis shows a distinct negative correlation between urbanization and ESH. The degree of negative correlation shows an upward trend with the processes of urban sprawl. The analysis results reveal the impact of urbanization on urban ESH and provide useful information for planners and environment managers to take measures to improve the health conditions of urban ecosystems.

## 1. Introduction

Urbanization is one of the main forces driving environmental and ecological change [1]. Characterized by population aggregation, urban expansion and economic development [2,3,4], urbanization has resulted in conversions of ecological land to constructed land, and as a result the conversions of natural ecosystems into human-dominated and coupled human-nature ecosystems [5,6,7]. Specifically, urbanization disrupts the flows of materials, energy and information and the structures and functions of ecosystems [8]. Nonetheless, the increasing population and urban development demand greater ecological services for the sustainability of social development [9], which gives rise to a mutually affecting relationship between urbanization and urban eco-environment [10]. Thus, monitoring the state of ecosystems and quantifying the effects of urbanization on urban ecosystems have become an important means for effective urban landscape planning and eco-environmental policy making.

The meaning of health in this context was extended from medical science to describe the state of regional ecosystems in the 1980s [11,12]. The change motivated a new concept, urban ecosystem health, which assists urban environment managers with integrating ecological, economic, social and human health factors and includes not only the health and integrity of natural and built environments, but also of urban residents and the whole society [13,14,15]. This research topic was driven by extensive public concern and decades of progress in ecosystem health research [16,17,18]. However, due to the complexity of ecosystems, it is challenging to develop a precise operational definition and find a uniform index system to evaluate their health conditions [19]. To address this issue, Costanza [20] provided the concept of ecosystem health (ESH), which was defined as the ability of an ecosystem to keep its original state and maintain its organizational structure. In addition, a healthy ecosystem can recover with its self-regulating processes after being disturbed by human activities [20]. ESH can be measured by using the indices related to three main aspects, i.e., vigor, resilience and organization [21]. These indices can assist environment managers to assess ecosystem states, and are conducive to the sustainable development of ecosystems. Vigor means that a system is active and has plenty of energy to maintain its health; organization refers to different components that exist between which some relationships can be found that which can make the system more stable and effective; and resilience describes whether a system can recover from interference and maintain its stable structure [22]. Adding to the notion of ESH by Costanza [20], Myneni [23] highlighted that ESH is comprehensive and multi-scale.

Recently, researchers have provided many methods for ESH assessment. For instance, landscape metrics were used to assess whether an ecosystem is health in a city landscape [13], and five indicators associated with ecosystem pressure and response were selected to describe estuary ecosystem health [24]. Nevertheless, among these methods, many studies have continued to use the framework of vigor-organization-resilience to investigate ecosystem health. Spatial correlations exhibited in these indices can exert crucial influences on ecological processes at the regional scale, which is the common scale at which ESH studies were conducted and environmental protection plans were made. However, in urban areas, some studies only selecting the indicators of vigor, organization and resilience when researching the regional ecosystem health. Specifically, a healthy regional ecosystem provides a range of valuable services sustainably, which is a primary design goal for ecological engineering benefit both humans and the entire natural world [25]. Thus a comprehensive method considering the four indicators of vigor, organization, resilience and ecosystem services were selected by researchers [25,26,27] to obtain better assessment results in various regions. 

Land use could also cause massive changes in ESH largely, and many studies have discussed the influence of the relationship between land use changes and ESH. Restoration projects such as the Grain to Green Program have made great contribution to solving environmental problems including soil erosion, flooding and desertification. Other researchers have investigated the influence of ecological protection on ESH by combining remote sensing images and geographic information system (GIS) techniques. However, most studies paid attention to the changes of ESH but fell short of assessing the effects of urbanization on urban ESH. For example, Liao et al. [26] assessed the relationship between changes in ESH and land use changes. Wang et al. [28] assessed the regional ecological health of Xiamen City, China. Both studies did not explore the effects of urbanization on ESH. Peng et al. [25] discussed the relationship between urbanization and changes of regional ESH levels in the early stage of the Chinese economic reform (1978–2005), but they did not consider the spatial patterns of regional ESH levels due to urbanization. In short, existing studies leave two gaps that require attention. First, there is a need to examine the clustering patterns between ESH and urbanization, especially at the local scale. Spatial correlations are commonly found between ESH and their drivers (including urbanization), leading to biases in the results obtained by ordinary least squares (OLS) and geographically weighted regression. Therefore, other statistical techniques dealing with spatial autocorrelation must be employed. Second, previous studies have focused on the regional scale, with administrative districts as the usual unit of spatial analysis. Analysis on this scale is underlain by a process aggregating locally collected data into meso-or macro-levels. This may limit the practical applicability of ESH in local urban landscape planning [29].

Additionally, remote sensing images are also instrumental in assessing and investigating ESH across space and time at temporal and spatial scales in an area [30,31]. For example, Liao et al. [22] built a pressure-state-response framework using remote sensing data and ecological service values in an assessment of regional ESH. Sun et al. [27] measured the ecosystem health of the Liaohe River Delta in China by combining remote sensing images, GIS technology and ecosystem services at the watershed scale.

The novelty of this study stems from the use of time series land use data from 1999, 2005, 2009 and 2013 and four main factors of vigor, organization, resilience and ecosystem services (ESV) to assess urban ESH. In addition, the clustering patterns between ESH and urbanization have been analyzed for their spatial auto correlative characteristics at the local scale. Compared with previous studies, of which the analyses were performed at the regional scale or using administrative districts, this study investigated the spatial relationships between ESH and urbanization at the local scale, which provides fine-grained recommendations for future urban ecological planning. Zhuhai City is studied because it has experienced continual and rapid urbanization, because it is designated as one of the special economic zones (SEZs) in China for its economic reform campaign in 1980 [32]. Since then, Zhuhai City has experienced more than 30 years of rapid urbanization, which has resulted in a dramatic decline in environmental quality and the destruction of natural landscape [33]. The specific aims of the study include the following: (1) assessing and quantifying the ESH and urbanization level in Zhuhai City using remote sensing images, gross domestic product (GDP) data and population data, (2) investigating spatial correlations between ESH and three kinds of urbanization and (3) assessing the spatial dependence of ESH on urbanization based on spatial regression models (SRMs), including spatial lag model (SLM) and spatial error model (SEM).

## 2. Materials and Methods

### 2.1. Study Area and Data Source

#### 2.1.1. Study Area

Zhuhai City is one of the major cities of Guangdong Province in southern China. It is situated on the west side of the Pearl River Estuary (Figure 1). The climate in Zhuhai is subtropical-monsoon-maritime. The urban land, population and economy experienced fast growth in the past three decades [32]. In 2013, the total land area of Zhuhai City was 1711 km^2^. The population increased more than fourfold, from 0.36 million in 1979 to 1.56 million in 2010. As a result, the economic structure in Zhuhai has undergone a fundamental shift. Agriculture and fisheries were the main industries in the early 1980s, but with the development of economy, transportation, high-tech, secondary and tertiary industries gradually made great contributions to the GDP, transforming Zhuhai into a modern city [34].

#### 2.1.2. Data Source

Land use maps were gathered from remote sensing images taken in 1999, 2005, 2009 and 2013, and used to extract normalized difference vegetation index (NDVI) to measure the values of vigor, organization, resilience and ecosystem services. Population and GDP data with a spatial resolution of 1 km × 1 km were obtained from the National Science and Technology Infrastructure of China (http://www.geodata.cn) [35].

The aforementioned sets of remote sensing imageries were also used to extract the land use land cover types needed for this study, which include farmland, developed land, wasteland, forestland, water body, grassland and unutilized land, through supervised classification. The kappa values of the classification results are all above 8.5, suggesting that the land use land cover maps developed are of sufficient accuracy for the purpose of assessing ESH.

### 2.2. Assessment of Ecosystem Health

In urban areas, it is well-known that there exists coupled relationships between human and natural systems. To assess the ESH of Zhuhai City, we used the following four indices: vigor, organization, resilience and ecosystem services (ESV). The ESH is evaluated with two variables, the physical health (PH), consisting of three indices, i.e., vigor, organization, resilience [20] and the ecosystem services (ESV) of Zhuhai City. The equations for calculating ESH are as follows:(1)$$H=\sqrt{\mathrm{PH}\times \mathrm{ESV},}$$
(2)$$\mathrm{PH}=\sqrt[{3}]{V\times O\times R},$$
where H is the ESH of the units being assessed, PH is the physical health of the ecosystem, ESV is the ecosystem services, and V, O and R refer to the vigor, organization and resilience of spatial entities.

To capture ecosystem vigor, which refers to the ecosystem’s metabolism or primary productivity, NDVI is used because it has been proven to be effective in assessing the primary productivity of vegetation [36,37]. The NDVI is calculated using Landsat satellite images with multispectral bands in different times [38,39]. Ecosystem organization refers to the structural stability of the ecosystems. It is acknowledged that spatial patterns are essential influencing factors in the management of ecosystem processes at a landscape scale [40]. In this study, ecosystem organization is determined by examining the spatial heterogeneity in landscape patterns, which can be measured through landscape diversity using Shannon’s diversity index (SHDI) and landscape fractal dimension using the area-weighted mean patch fractal dimension (AWMPFD) [41]. In this study, the formula used to calculate the ecosystem organization is as follows:(3)O=0.7×SHDI+0.3×AWMPFD,
where O is ecosystem organization of the spatial entities, SHDI is Shannon’s diversity index, AWMPFD is the index of area-weighted mean patch fractal dimension, the calculation was run using Fragstats 4.2.

Resilience is the ability of a natural ecosystem to recover to its original structure and functions after external disturbance. The formula used to calculate ecosystem resilience is as follows [25]:(4)$$\mathrm{ER}=\sum_{i=1}^{n}A_{i}\times R_{i},$$
where ER is the ecosystem resilience of spatial entities, A*_i_* is the area ratio of land use type *i*; R*_i_* is the ecosystem resilience coefficient of land use type *i* and *n* is the number of land use types. In order to generate the comparable study results, the values of ESH in four years studied will be divided into the following five levels: well (80–100), relatively well (60–80), ordinary (40–60), relatively weak (20–40) and weak (0–20).

### 2.3. Quantifying Ecosystem Services

This study quantified the ecosystem services of Zhuhai by following the approaches in Costanza et al. [42] and Xie et al. [43]. We extracted the market price of cereals punished in The Guangdong Statistical year books of 1999, 2005, 2009 and 2013 available from the National Library of China official website (http://www.nlc.cn) [44], and calculated the average ESV of one equivalent value for Zhuhai City, which was 1539.02 yuan/(ha*a). Using the area ratio of different land use types, the total ecosystem values of each grid in the study area were derived.

### 2.4. Mapping Urbanization Levels

Urbanization levels can be investigated from several respects: (1) population growth, which is the main feature of the development of a modern city; (2) economic development, which is the main means of urban development; (3) the expansion of constructed land, which links directly to population growth and economic development and (4) living standards improvement, which is the result of urbanization [45]. As it is difficult to obtain accurate socioeconomic data that measures the improvement of living standards, let alone assesses their spatial heterogeneity, this study used only the first three kinds of data to represent urbanization levels: the population growth by population density (POPD; person km^−2^), economic development by density of gross domestic product (GDPD; 10^4^ yuan km^−2^) and expansion of constructed land by constructed area proportion (CAP) [46,47,48].

### 2.5. Spatial Correlation Test

The relationship between ESH and urbanization can be investigated by using bivariate Moran’s I (Figure 2). In our study, we explored the spatial correlation from two aspects, spatial clustering (positive spatial correlation) and spatial dispersion (negative spatial correlation), based on global bivariate Moran’s I and local bivariate Moran’s I (bivariate local indicators of spatial association (LISA)). Global bivariate Moran’s I mainly focus on investigating spatial correlations between ESH and urbanization on large scale or across the entire study areas such as Zhuhai City, while local bivariate Moran’s I prefer to evaluate the spatial correlations within different spatial units, such as grids in this study [49]. The formulas used to calculate Moran’s I are as follows:(5)$${E\prime }_{i.j}=\frac{E_{i.j}-E_{i.\min}}{E_{i.\max}-E_{i.\min}},$$
(6)$$I_{\mathrm{eu}}=\frac{N\sum_{i}^{N}\sum_{j\neq i}^{N}W_{\mathrm{ij}}Z_{i}^{e}Z_{j}^{u}}{\left(N-1\right)\sum_{i}^{N}\sum_{j\neq i}^{N}W_{\mathrm{ij}}},$$
(7)$${I\prime }_{\mathrm{eu}}=Z^{e}\sum_{j=1}^{N}W_{\mathrm{ij}}Z_{\mathrm{ij},}^{u}$$
where I_eu_ and I’_eu_ refer to the bivariate Moran’s I for ESH and different urbanization levels, respectively; N refers to the number of grids; w_ij_ refers to the spatial weight matrix for calculating spatial correlations between pairs of adjacent spatial units, which was generated based on queen contiguity weight with the first order neighbor in a 3 × 3 matrix [50,51] and z_j_^e^ and z_j_^u^ refer to the standardized value of ESH and urbanization indicators (GDPD, CAP and POPD) of each spatial grid using Equation (5). The values of I_eu_/I′_eu_ range from –1 to 1, indicating the neighboring grid cells have distinctive and similar values, respectively. In addition, they provide measures of the magnitude urbanization influences ESH. The larger the value is, the greater the impact of urbanization on ESH. The study carried out 999 permutations to test the significance (*p* < 0.05) in this context [52].

Bivariate LISA produces graphical outputs including Moran scatter plots, cluster maps and corresponding significance maps to help users visualize local spatial correlations. They illustrate the relationship between the value of ESH at a given location and the average value of urbanization level at neighboring locations at a certain significance level. The four quadrants of a cluster map thus generated represent four types of local spatial autocorrelation: quadrant I (high–high type, HH) indicates high ESH values surrounded by high urbanization values; quadrant II (high–low type, HL) indicates high ESH values surrounded by low urbanization values; quadrant III (low–high type, LH) indicates low ESH values surrounded by high urbanization values and quadrant IV (low–low type, LL) indicates low ESH values surrounded by low urbanization values.

## 3. Results

### 3.1. Assessment of Urban Ecosystem Health

Figure 3 shows the results of ESH values obtained for the four years studied using the five levels described in Section 2.2. Overall, the areas categorized as weak and relatively weak increased from 1999 to 2013. More specifically, between 1999 and 2005, the proportion of the study area characterized as relatively well or better was well above 50%. After 2005, however, the ESH of the study area deteriorated. The proportion of total areas categorized as relatively well and well dropped below 50%, to 45.82% in 2013, an indication that the urban ecosystem had gradually become unhealthy with the progress of urbanization.

The spatial patterns of the ESH levels of Zhuhai City from 1999 to 2013 are shown in Figure 4. In 1999, the areas categorized as weak were mainly distributed in Meihua, Jida and Gongbei Town, which are the main urban areas of Zhuhai City. They were also found at the border of Jingan Town and Baijiao Town, and the aviation industry park area in the south of Zhuhai City. The areas categorized as relatively weak were mainly distributed in Doumen Town, Ganwu Town, Pingsha Town, Hongqi Town, North of Hongqi Town and Baiteng streets. The areas categorized as ordinary were scattered throughout the region of Zhuhai City, mainly around the areas of relatively weak health. The areas categorized as relatively well and well were mainly distributed in Jingan, Doumen and the border of Ganwu Town where several forest parks in these areas, such as Baizu Mountain, Zhuzi Ridge and Xinping Mountain are located. In addition, Nanshui Town in the south of Zhuhai City, Sanzao Town and the southern part of Hengqin Town also belong to relatively well and well areas. As far as the eastern part of Zhuhai City, the areas categorized as relatively well include: the College Town, the bonded area and the Nanping Science and Technology Park. These areas are distributed in the mountain areas and forest reserves in Zhuhai, so the environment in these areas was well protected by the city government of Zhuhai.

In 2005, the ESH in the northeastern part of the study area had deteriorated given that the weak ESH areas level had expanded. The health states of the ordinary weak or relatively weak were further reduced. For example, the areas around Baizu Mountain in the west part of Zhuhai City experienced a decline in ESH levels in this period and the border areas with well and relatively well health of Nanping Town, Free Trade Zone and Hengqin Town also decreased into the lower ESH.

In 2009, the most obvious characteristic of urban ESH was that the areas with weak health level drastically increased, especially the areas with relatively weak health level in 2005. This result was the same as that in Figure 3, which shows that the proportion of weak health areas increased significantly. For instance, areas categorized as weak health in the northern part of College Town, the junction of Hengqin Town and the bonded area, as well as Doumen Town, Ganwu Town, Jing’an Town and Sanzhao Town increased significantly.

Compared to other years, areas with a weak health level in 2013 were the highest, and almost all the relatively weak areas in 1999 deteriorated to weak level in 2013. The same trend was also found in other levels of ESH, e.g., the areas in the north part of the university campus, Baizu Mountains and northern areas of Nanshui Town deteriorated to weak level of health from 2009 to 2013. Overall, in 2013, only 45.82% of areas throughout Zhuhai City had a well and relatively well healthy urban ecosystem. The results suggested that it is necessary to take active protective measures in the development of Zhuhai City and impose effective regulations to protect the well and relatively well areas.

### 3.2. Spatial Distribution of Three Kinds of Urbanization Level 

In order to verify the relationships between rapid urbanization and urban ecological health, the study mapped the GDP density (GDPD), constructed area percentage (CAP) and population density (POPD) of Zhuhai City during the study period (Figure 5). It could be found that the spatial distribution of GDPD, CAP and POPD carry similar patterns and temporal trends with the progress of urbanization. The area with the highest level of urbanization was in the city center, gradually decreasing from the city center to peripheral areas. The economic urbanization as described by GDPD was highest in the eastern part of the study area, especially in the main city area, in 1999, 2005, 2009 and 2013. In addition, the economic urbanization level in the southern part of the city also drastically increased during 2009–2013. Land urbanization, which is represented by CAP, was the highest in the main city and gradually decreased to the lowest at the periphery in 1999. While similar in the spatial patterns, differences between economic and land urbanization are obvious. The areas with high land urbanization levels could be found around city parks (e.g., Bailiandong Park, Marina Park and City Park) and ecological conservation zone (e.g., Banzhang Mountain, Shijing mountain and Shihua Mountain) with the development of the city. It seems clear that the constructed area in the study area formed several centers around Jinan Town, Sanzao Town and Doumen Town instead of one center within Zhuhai City. In addition, many areas exhibited sudden drops of constructed area percentage. As for population density, the highest level was found only in Zhuhai City Center. This shows that economic development and the immigration of the population occurred later than urban land expansion in the progress of urban urbanization. It also suggests the need to pay more attention to urban sprawl in the study area in case of possible ecosystem damage.

### 3.3. Effect of Urbanization on Urban Ecosystem Health

Table 1 shows the results of Moran’s I analysis of GDP density (GDPD), population density (POPD) and constructed area proportion (CAP). All results are below 0, suggesting that urbanization measured by the three indicators had a negative impact on ESH. It can be inferred that the areas with low ESH values may be surrounded by or adjacent to areas with high urbanization. Additionally, different kind of urbanization exerted various negative effects across years.

In 1999, the largest negative impact of urbanization was found between land urbanization and ESH, indicating that urban expansion had a stronger impact than the other two types of urbanization. In 2005, a negative correlation between the three kinds of urbanization and ESH showed a similar trend as in 1999, but the degree of negative correlation between population and economic urbanization visibly increased, indicating their increased influence on the deterioration of ESH. In 2009 and 2013, it could be found that the degree of negative correlation between all kinds of urbanization and ESH continuously increased with the development of the city.

Figure 6 shows the spatial correlations between ESH and urbanization levels. The figure reveals similar clustering patterns of spatial correlation. The HH areas for GDPD and ESH were mainly concentrated in the northern and southern parts of the main areas of Zhuhai City in 1999. With the development of the urban economy, the size of HH areas decreased significantly especially, in the period 2009–2013. The HH areas for CAP and ESH only accounted for a small proportion in four years, and they were scattered across the whole study area. In 2013, the HH areas occupied only 3.08% of the total study area. The HH areas for POPD and ESH showed a similar trend with that in HH areas for GDPD and ESH from 1999 to 2013. The proportions of the kind of areas drastically decreased with the increased population. 

The HL areas for GDPD and ESH were mainly distributed in the western part of Zhuhai and gradually deceased from 1999 to 2013. They occupied 20.34% of the total study area. The HL areas for CAP and ESH were scattered in the southern part in Zhuhai City in 1999 and 2005, then clustered in the southern part. The HL areas for POPD and ESH were mainly concentrated in northeastern Zhuhai in 1999, then decreased with the increased population in 2005, 2009 and 2013. 

The LL areas for GDPD and ESH were distributed in the northwestern part of Zhuhai City, then decreased along with GDPD growth during the period 1999–2013. In 2013, the LL areas for GDPD and ESH occupied about 7.76% of the total study area. The LL areas for CAP and ESH were scattered across the whole study area, but decreased with the processes of the urban expansion. In addition, the LL areas only accounted for 2.82% of the total area. The LL areas for POPD and ESH were mainly distributed across the western part of the study areas and other LL areas were clustered in the southwestern part of Zhuhai City in 1999. Over time, they showed a decreasing trend with the increasing population from 2005–2013.

The LH regions for GDPD and ESH were concentrated in the major city and northeastern part of the study area throughout the study period. They then decreased with GDPD growth. By 2013, the LH regions occupied 6.78% of the total study area. The LH areas for CAP and ESH were concentrated in the main city across the whole study period. At the same time, some areas in western Zhuhai belonged to this kind of area. The LH areas for POPD and ESH showed the same trend as GDPD and ESH. They also concentrated in northeastern Zhuhai City in 2009. In 2013, they occupied 6.41% of the total study area.

### 3.4. Spatial Dependence of ESH on Urbanization

The results of ordinary least squares (OLS) regressions, which are presented in Table 2, show spatial dependencies in all regressions. In addition, the spatial dependencies were more significant between the LM lag and LM error than OLS for all ES regressions, so SEM was used for GDP density (GDPD), population density (POPD) and constructed area proportion (CAP) regression in this study.

The results of R^2^ and log likelihood in spatial regression were higher in spatial regression models (SRMs) than OLS, and the Akaike information criterion (AIC) and Schwartz criterion (SC) values were lower in SRMs than OLS, which indicated that the results in SRMs are more reliable than that in OLS for all regressions (Table 2 and Table 3). In order to obtain the relative effect of every index, we analyzed the regression coefficients. The results of the error coefficient (lambda) were significantly positive (*p* < 0.01) in SEM for all regressions, indicating that the non-urbanization factors also exert positive influences on all three indicators. The coefficients between IUL and all ESH indices showed that urbanization resulted in the decline of ESH. In addition, there was an increasing trend in the absolute values of CUL coefficients between 1999 and 2005 and between 2005 and 2009, implying that the impact of urban expansion on ESH is becoming more pronounced.

## 4. Discussion

### 4.1. ESH in Zhuhai City Changed from 1999 to 2013

In this study, ESH in Zhuhai City was first assessed in 1999, 2005, 2009 and 2013, then the assessment results were divided into five levels: weak, relatively weak, ordinary, relatively well and well. The changes in areas corresponding to different levels from 1999 to 2013 showed that the overall quality of urban ecosystem health decreased during the period of study, indicating that urban planners and environment managers should pay more attention to protecting ESH. Additionally, Zhuhai has experienced rapid urbanization for 30 years since the commencement of the economic reform in China. The expanded constructed area and rapidly increasing population have caused an obvious decline in ESH [53]. These results are similar to studies in other rapidly developing cities in China such as Shanghai [46], Shenzhen [25] and Beijing [54]. From the temporal perspective, the results showed a clear declining trend in ecosystem conditions from 1999 to 2013, as the areas with weak health level increased by nearly six times during this period. On the other hand, relatively weak areas showed a declining trend rather than a dramatic increase, the same trend was found for ordinary areas, and these two kinds of ESH levels accounted for the relatively low percentage in all four years. Additionally, well and relatively well areas showed a slightly decreasing trend and the proportion of them decreased about 10% of the total study area, this phenomenon indicates that the whole study area, including the four levels of ESH, was on a downward trend instead of a specific class of ESH converting to a weak health level ecological system. It signals the urgent need to protect the whole urban ecosystem rather than only pay attention to weak and relatively weak regions.

In terms of spatial dimensions, we evaluated the spatial distribution of urban ecological system health for all four years. The results show that the weak areas mainly located in the city center, urban areas and constructed areas, and these kinds of areas constantly increased from 1999 to 2013. This implies that Zhuhai paid more attention to economic development and urban expansion during the study period without considering ESH protection. In areas with less human activity, such as mountains and nature reserves, existing studies have shown that ESH always remained at the relatively high level [46,55]. In Zhuhai City, there are many mountains and much forest land distributed in areas such as College Town and DouMen Town with low POPD, which also plays a certain role in ESH protection, indicating that the Zhuhai government carried out related management of the ecosystem of these regions [32]. The relatively weak and ordinary areas mainly around urban areas or the city center correspond to weak level areas, therefore regions around or adjacent to urban centers need to pay attention to improving and protecting the urban ecosystem when enjoying the economic expansion of the city center.

### 4.2. Spatial Spillover Effect in the Relationship between ESH and Urbanization

In order to obtain a more complete understanding of the impact of urbanization on ESH, this study measured the spatial relationship between ecosystem health and urbanization. Three indicators GDPD, CAP and POPD were selected as explanatory variables to represent the main characteristics and feature of urban development, while ESH was selected as the dependent variable. In addition, this study investigated the impact of urbanization on ESH from the aspects of population growth, economic development and urban expansion. The results showed that the states of ESH was affected by the urbanization level in its neighbor areas, thus the existence of the spillover effect, which refers to a spatial externality resulting from place-based proximity, i.e., one unit enjoying benefits or incurring costs from its neighbors [29]. In this study, a decreased level of ESH in some regions may result from an increasing level of urbanization in neighboring areas according to the results in Section 3.3. A possible reason for this may be that urban areas continue to exchange energy and materials with neighboring areas. Changes of environmental elements (e.g., precipitation, temperature and carbon dioxide) in a given location are likely to propagate to surrounding areas through natural processes such as atmospheric movement and animal migration, of the ability of the surrounding areas to provide ESs [46,56]. This helps us to understand why not only urban centers, but also suburban areas, were observed to have large areas of low ESH and high urbanization.

Bivariate LISA (Figure 5) and the results in Table 3 show that it was not always urbanization that exerted a negative impact on ESH at the local level, indicating that researchers should further explore the relationship between ESH and urbanization because there may exist other factors such as vegetation types and roads that could contribute to changes in ESH.

### 4.3. Better Management of the Urban Ecological Environment by Incorporating the Spatial Relationship between ESH and Urbanization

Urban planners and environment managers are always faced with the question of how to balance the relationships among urban construction, economic development and ecological conservation. Although some solutions, such as demarcating ecological protection areas, were suggested by many studies, there are some problems. For instance, some research did not consider the influences of urbanization on environmental management. To address this concern, this study provides several recommendations for better environmentally friendly urban planning.

Four types of clustering patterns (i.e., HH, HL, LL and LH) between three variables that measure urbanization, i.e., population, economy and constructed area urbanization and ecosystem health were explored in this study. Urban planners should set areas with high ESH as ecologically friendly land use types. For instance, the urban green space including parks, forests and grass land should be classified as this kind of land use type. Areas with a high urbanization level should be given more attention to protect them from further deterioration Measures such as developing more green spaces or converting constructed areas into urban green space should be adopted. Areas with high ESH and high urbanization should be recognized as “ecological function regions”, in which the environmental friendly land use type could be encouraged. Areas with low ESH and low urbanization should be considered to give priority to building urban green spaces.

## 5. Conclusions

This study investigated spatiotemporal changes of urban ecosystem health and three indicators of urbanization (GDPD, CAP and POPD) in 1999, 2005, 2009 and 2013 at the urban scale. By combining remote sensing analysis, which derived CAP and ESH data, and statistical data published by the National Science and Technology Infrastructure of China, the study measured the spatial correlations between the three indicators and ESH in order to gain a better understanding of how urban ecosystems can be protected. As urban ecosystems are complex and open, they are very susceptible to the surrounding environment. Thus it is imperative to explicitly consider spatial dependencies between ESH and urbanization to better characterize urban ecosystems. The results of this study support the following conclusions. First, ESH was negatively correlated with all three types of urbanization, i.e., economic, constructed area and population urbanizations when measured for the whole study area. Nonetheless, it was found that there were four distinct kinds of patterns of local correlations according to the bivariate LISA method. The results showed that different management approaches could be developed according to the characteristics of different regions. Second, this study discovered a spillover effect in relationships between ESH and urbanization. The results of this study on the relationship between urban development level and ESH can provide practical guidance for future urban environmental protection and construction.

## Figures and Tables

**Figure 1 ijerph-16-04717-f001:**
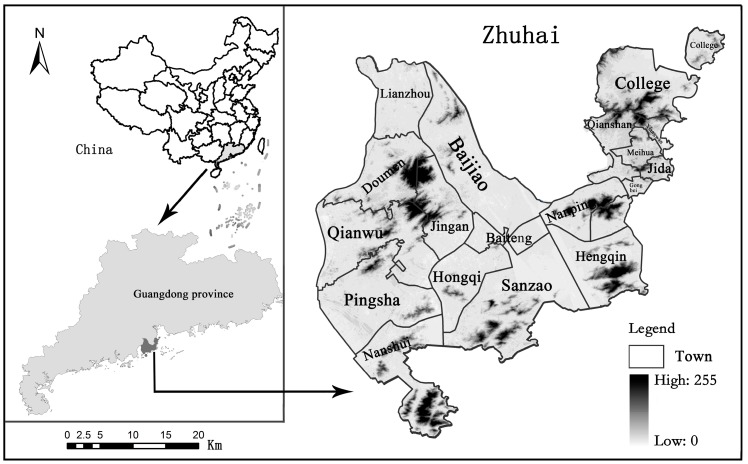
Location of Zhuhai City.

**Figure 2 ijerph-16-04717-f002:**
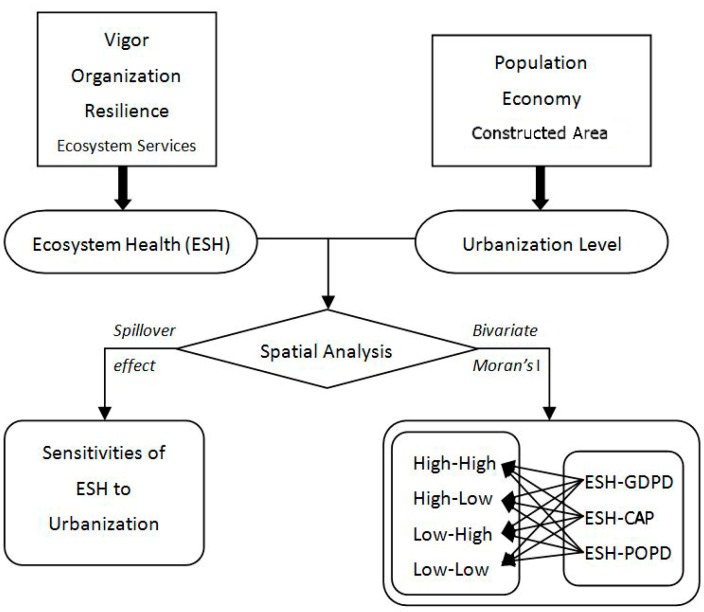
Assessment of spatial correlation between urbanization and ecosystem health (ESH). GDPD, density of gross domestic product; CAP, constructed area proportion; POPD, population density.

**Figure 3 ijerph-16-04717-f003:**
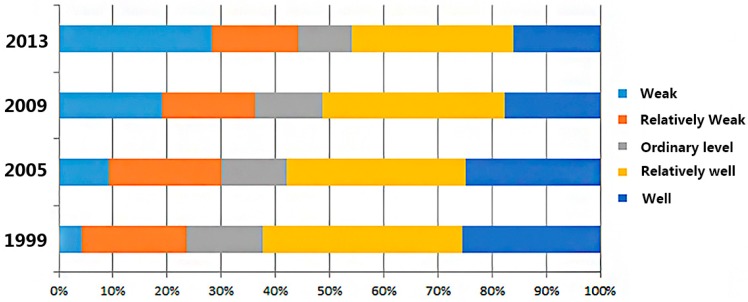
With different ecosystem health levels in 1999, 2005, 2009 and 2013.

**Figure 4 ijerph-16-04717-f004:**
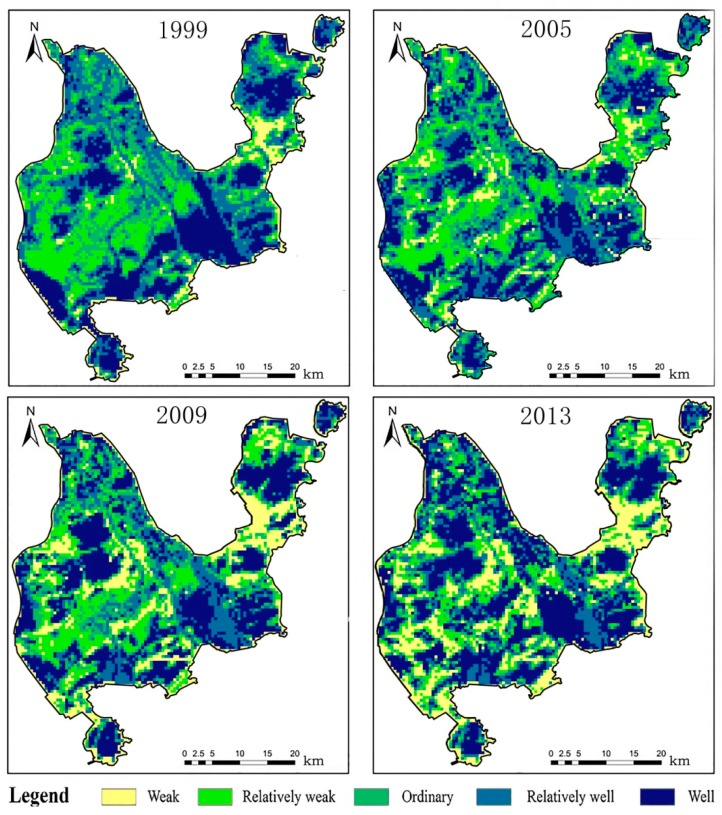
Spatial patterns of urban ecosystem health in 1999, 2005, 2009 and 2013.

**Figure 5 ijerph-16-04717-f005:**
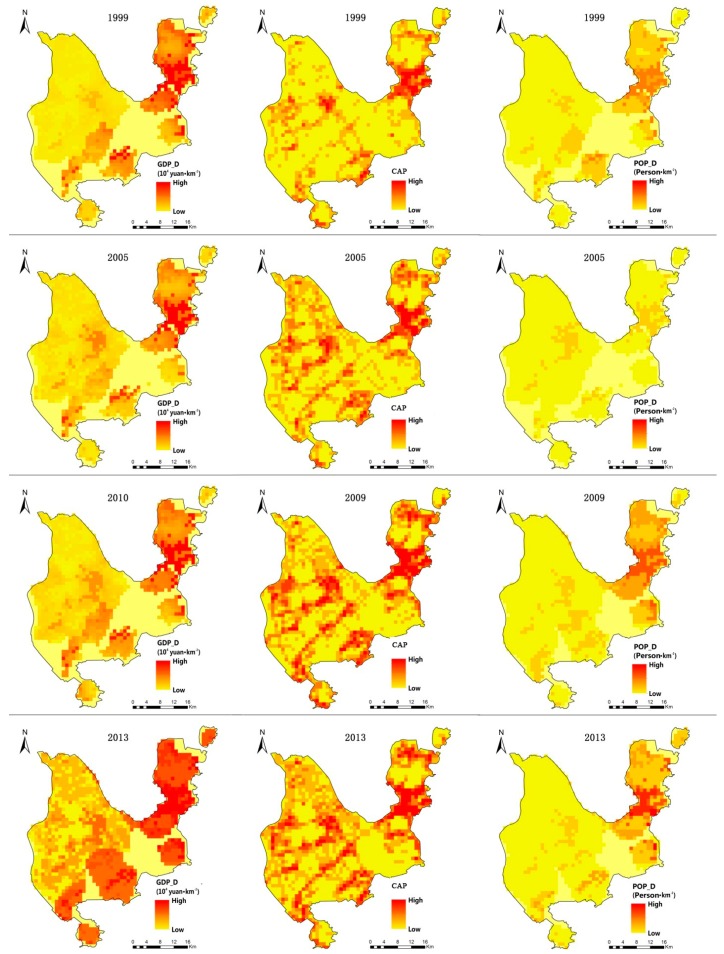
Urbanization levels in Zhuhai City. GDPD, GDP density; CAP, constructed area proportion; POPD, population density.

**Figure 6 ijerph-16-04717-f006:**
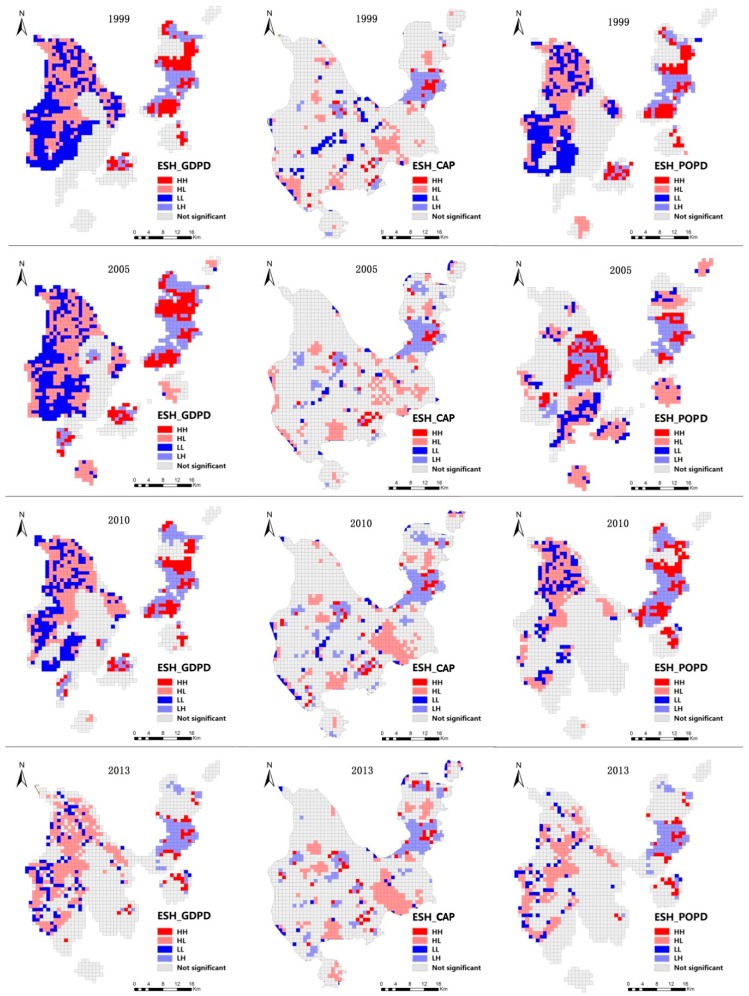
Local indicators of spatial association (LISA) cluster maps between ESH and individual urbanization level (IUL; GDPD: GDP density; CAP: constructed areaproportion; POPD: population density; HH: high ESH and high IUL; HL: high ESH and low IUL; LH: low ESH and high IUL; LL: low ESH and low IUL).

**Table 1 ijerph-16-04717-t001:** Moran’s I between ESH and GDPD, POPD and CAP.

IUL	Year	1999	2005	2009	2013
GDPD	Moran’s I	–0.0484	–0.1198	–0.1477	–0.2463
	z-Value	–4.107	–19.84	–12.66	–15.64
	*p*-Value	0.001	0.001	0.001	0.001
POPD	Moran’s I	–0.0883	–0.1654	–0.1255	–0.2619
	z-Value	–7.516	–26.04	–8.322	–16.49
	*p*-Value	0.001	0.001	0.001	0.001
CAP	Moran’s I	–0.2039	–0.2001	–0.2738	–0.257
	z-Value	–15.99	–15.87	–21.18	–19.66
	*p*-Value	0.001	0.001	0.001	0.001

**Table 2 ijerph-16-04717-t002:** Results of ordinary least squares (OLS) regressions between ESH and IUL.

Dependent	ESH1999	ESH2005	ESH2009	ESH2013
Constant	0.50621	0.620177	0.664083 **	–82.5178
GDPD	0.195904 **	0.148384	0.120648 *	–159.412
POPD	–0.00491	–0.159545	0.168452 **	–438.546
CAP	–0.757281 **	–0.656092	–0.945341 **	173.391
R^2^	0.199883	0.242939	0.389663	0.005089
Log likelihood	165.233	176.358	–34.5155	–11255.2
AIC	–322.465	–344.716	77.0309	22518.5
SC	302.395	–324.53	96.992	22539.3
Moran’s I	20.7746 **	11.9284 **	18.0845 **	9.1422 **
Lagrange multiplier (lag)	367.6217 **	127.6153 **	262.6840 **	79.0298 **
Robust LM (lag)	1.0614	2.51	6.4710 *	0.1599
Lagrange multiplier (error)	419.5618 **	136.4934 **	316.9356 **	79.6533 **
Robust LM (error)	53.0015	11.3888 **	60.7225 **	0.7833

GDPD, GDP urbanization; CAP, constructed area proportion; POPD, population urbanization; AIC, Akaike information criterion; SC, Schwarz criterion. * *p*-values at 5% level. ** *p*-values at 1% level.

**Table 3 ijerph-16-04717-t003:** Results of spatial regressions between ESH and IUL.

Dependent variables	ESH1999	ESH2005	ESH2009	ESH2013
LAMBDA	0.613031 **	0.420321 **	0.563871 **	0.425095 **
Constant	0.517883	0.611472 **	0.660955 **	–110.963
GDPD	0.113996 **	0.133592 **	0.094901	–130.042
POPD	–0.00579	–0.148248 **	0.140503 *	–557.4
CAP	–0.747333 **	–0.626769 **	–0.888831 **	250.482
R^2^	0.415862	0.325404	0.524507	0.089752
Log likelihood	300.803312	225.35109	69.3103	–11215.7
AIC	–593.607	–422.702	–130.621	22439.3
SC	–573.537	–422.516	–110.66	22460.1

AIC denotes Akaike information criterion. SC denotes Schwarz criterion. LAMBDA denotes spatial error term of ESH in 1999, 2005, 2009 and 2013. * The values of Pat 5% level. ** The values of Pat 1% level.

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
