# Peer review of "Impact of Urbanization on Ecosystem Health: A Case Study in Zhuhai, China"

_ijerph, 2019, doi:10.3390/ijerph16234717_

Round 1

Reviewer 1 Report

A new approach to assess the urban ecosystem health is proposed; A tool GIS is used to process image data.

In the introduction section a brief description of the proposed approaches in literature appied in order to assessthe urban ecosystem health it is necessary. The authors must highlight what are the performance advantages of the proposed approach compared to other methods applied to asses the urban ecosystem health.

It needs to add a schematization  of the  proposed framework and the processes used that highlight the sequentiality.

The process described in paragraph 2.5 which uses the T global bivariate Moran's I and local bivariate Moran's I to analyze spatial clustering and spatial dispersion is not very clear and should be described in more detail.

Reviewer 2 Report

In its present form, the subject of the paper only very loosely refers to the main objectives of the IJERPH journal.

The work is of international interest; however, it is mainly of interest to scietists from Asia.

In my opinion, the manuscript presented does not refer to International Environmental Research and Public Health. This topic is treated only briefly. I have the impression that the authors use the term "ecosystem heath" to merely show the existence of the word "health" in the name of the magazine. There is no reference to "public health" at all in the manuscript. The authors have already published an article on similar topics in the journal Sustainability. In this case, using the same methods and the same research area, they try to show that they have created a completely new quality. In addition, China's poevinence is not a representative area for the whole world, so it limits its scope of influence only to Asian countries. I will not mention that the manuscript contains numerous editorial errors, e.g. repeated keywords, incorrect way of citing literature, no spaces, different fonts appearing in the text, etc. The manuscript was not prepared carefully. The discussion in both forms is not a discussion - there are no references to literature data, including those on international scope. In both forms it is rather a summary of the results obtained in the manuscript, rather than a discussion.
I am not saying that the reviewed manuscript has no scientific value. However, I am saying that he does not fit into the subject of the journal.

Reviewer 3 Report

The study certainly has some potential to be published but in the current form it needs a substantial revision of all the manuscript sections that currently lacks practical and important information to clearly understand the results. In particular, in the sub-section “The Assessment of Ecosystem Health”, I did not understand which data were used to assess the four indices “vigor, organization, resilience and ecosystem services (ESV)”. What is the data source? What is the temporal and spatial format resolution? Many basic information is lacking to understand what data have been used. In the sub-section “Quantifying ecosystem services” I could not access to the page http://www.zhzgj.gov.cn/, is it working? Unfortunately, at the moment too many details are missing to understand the approach adopted and the data used. It is very important to mention and clearly describe the data source. In addition, throughout the manuscript there are too many words attached that often confuse the reader. Please check. In the “Results” section, several important information should be included and clearly explained in the “Materials and Methods” section. In particular, I refer to the paragraph concerning the lines between 144 and 149. I would like to understand the choice of classification adopted on what basis it is based. Is it an arbitrary choice or is it based on other criteria?
I suggest a significant and important revision and reorganization of the work, providing useful details so that the work (which as I said at the beginning has potential to be published, but not in this form) is clearly understandable and useful for a reader.

Round 2

Reviewer 2 Report

In the second version of the manuscript, the authors were trying to modify the work as much as possible with additional elements suggested by reviewers. Personally, however, I think that the explanations given, including the explanation of the method used, are still not so comprehensive. This hinders or even prevents repetition of work, and thus comparison with other locations. In my opinion, this is definitely the main limitation of the presented research.

There are still many situations in which editorial corrections are needed, e.g. there are no spaces between two consecutive words, in the References chapter different font types are used. Citations are also missing DOI numbers. In addition, I am not sure if lines 33 and 36 are certainly cited in the same literature. In line 33 there should be definitely more literature than one item. In addition, the full names of the abbreviations used should be included in the Results subsections.

Reviewer 3 Report

Surely the authors have tried, in the little time available for the revision, to modify as much as possible the work that now contains some extra elements to be understood. Personally, however, I believe that the explanation of the method adopted is not so exhaustive as to easily allow the repeatability of the work and therefore a comparison with other locations. This certainly represents the main limitation.

There are still many situations where there is no space between two consecutive words (I don't know if it is a display problem on the pdf or if it is a formatting error).

Minor comments:

Line 19: please check the sentence “… and socio-economic data in from 1999, …”

Line 19 and 115: a space between words is missing “(Thematic Mapper)TM images”

Line 117: A capital letter is missing “…landscape [34]. the specific…”

Line 119: please check the sentence “Using to the area ratio of different …”

Lines 230-233: This concept should be included in materials and methods section.

Line 317: : a space between words is missing “…urbanization asmeasured…”
